# Prohexadione-Calcium Reduced Stem and Tiller Damage and Maintained Yield by Improving the Photosynthetic and Antioxidant Capacity of Rice (*Oryza sativa* L.) Under NaCl Stress

**DOI:** 10.3390/plants14020188

**Published:** 2025-01-11

**Authors:** Wanqi Mei, Shaoxia Yang, Jian Xiong, Aaqil Khan, Liming Zhao, Xiaole Du, Jingxin Huo, Hang Zhou, Zhiyuan Sun, Xiaohui Yang, Nana Yue, Naijie Feng, Dianfeng Zheng

**Affiliations:** 1College of Coastal Agriculture Sciences, Guangdong Ocean University, Zhanjiang 524088, China; 2112204002@stu.gdou.edu.cn (W.M.); yangxx@gdou.edu.cn (S.Y.); 2112204010@stu.gdou.edu.cn (J.X.); aaqil_agron@hotmail.com (A.K.); nkzlm@126.com (L.Z.); duxiaole2022@163.com (X.D.); 2112204014@stu.gdou.edu.cn (J.H.); zjh1798@163.com (H.Z.); byndszy@126.com (Z.S.); xiaohuiyang99@163.com (X.Y.); 15873696981@stu.gdou.edu.cn (N.Y.); 2South China Center of National Saline-Tolerant Rice Technology Innovation Center, Zhanjiang 524088, China

**Keywords:** rice, Pro-Ca, main stem, tiller, NaCl stress, yield

## Abstract

Salt stress is a vital environmental stress that severely limits plant growth and productivity. Prohexadione-calcium (Pro-Ca) has been extensively studied to regulate plant growth, development, and stress responses. However, the constructive role of Pro-Ca in alleviating damages and enhancing rice tillers’ morph-physiological characteristics under salt stress remains largely unknown. The results showed that Pro-Ca significantly improved Changmaogu’s (CMG’s) productive tillering rate and the total yield per plant by 17.1% and 59.4%, respectively. At tillering stage, the results showed that Pro-Ca significantly improved the morph-physiological traits, i.e., leaf area, and photosynthetic traits of the rice variety with salt tolerance, under NaCl stress. Pro-Ca significantly increased the seedling index of the main stem and tiller by 10.3% and 20.0%, respectively. Pro-Ca significantly increased the chlorophyll *a* (chl *a*), chlorophyll *b* (chl *b*) and carotenoid contents by 32.8%, 58.4%, and 33.2%, respectively under NaCl stress. Moreover, Pro-Ca significantly enhanced the net photosynthetic rate (*A*) by 25.0% and the non-photochemical (NPQ) by 9.0% under NaCl stress. Furthermore, the application of Pro-Ca increased the activities of antioxidant enzymes by 7.5% and 14.7% in superoxide dismutase (SOD), 6.76% and 18.0% in peroxidase (POD), 26.4% and 58.5% in catalase (CAT), 11.0% and 15.9% in ascorbate peroxidase (APX), and Pro-Ca reduced the membrane damage index by 10.8% and 2.19% in malondialdehyde (MDA) content, respectively, for main stem and tiller leaves under NaCl stress. Pro-Ca significantly enhanced the soluble protein content of the main stem and tiller leaves by 2.60% and 6.08%, respectively. The current findings strongly suggested that exogenous application of Pro-Ca effectively alleviated the adverse impact of NaCl stress on the main stem and tillers by enhancing the photosynthetic capacity and antioxidant enzyme activity, and ultimately increased the productive tillering rate and grain yield.

## 1. Introduction

With the growth of the world’s population, it is estimated that food production will need to increase by 87 to 100 percent by 2050 to meet food demand. This largely requires an increase in the yield of food crops [1,2]. The main sources of salt accumulation in the soil tillage layer are irrigation water and seawater containing trace amounts of NaCl [3]. Satellite images show that the sea level continues to rise at a rate of about 0.34 cm per year, which has a negative impact on the quality of water and land near the ocean [4]. Salt stress is the most serious and widespread abiotic stress, affecting agricultural productivity in more than 20% of the world’s arable land [5]. Rice (*Oryza sativa* L.) is an important cereal crop for half of the world’s population [6]. It is a moderately salt-sensitive plant. Rice tillering determines the number of rice panicles, and panicle number is an important part of yield [7]. Changmaogu (CMG), a seawater-adapted rice landrace [8], exhibits far superior salt tolerance compared to even the popular salt-tolerant rice cultivar Pokkali [9]. High salt concentration will lead to a decrease in the number of tillers per plant, a decrease in biomass, a decrease in plant height, and a decrease in 1000-grain weight, which will eventually lead to a decrease in grain yield [10,11,12]. Therefore, improving the salt tolerance of crops is not only conducive to the development and utilization of saline–alkali land but also of great significance for ensuring world food security.

Salt stress mainly affects plants through ion toxicity and osmotic stress. Ion toxicity can affect the absorption of necessary nutrients such as potassium and calcium, accumulate harmful ions, and cause ion imbalance. A high salt environment hinders the absorption of water by plant roots and affects cell water potential [13]. Ion toxicity and osmotic stress can seriously affect the photosynthetic capacity of plants, mainly by inhibiting the biosynthesis of chlorophyll [14], increasing stomatal closure leading to reduced carbon dioxide supply [15], changing enzyme activity [16], and enhancing non-photochemical quenching for light protection [17]. The low photosynthetic capacity under salt stress leads to the imbalance of light energy absorption and utilization in the process of carbon dioxide assimilation, releasing more electrons and inducing the production of reactive oxygen species [15]. The production of reactive oxygen species inhibits the synthesis of proteins required for the repair of photosystem II (PSII) damage [18]. To mitigate salt damage, plants implement three strategies to improve tolerance: (A) restore ion homeostasis; (B) restore osmotic homeostasis; and (C) prevent and repair cell damage. Ion homeostasis mediated by membrane ion transporters is the main salt tolerance response mechanism [19]. Osmotic homeostasis and cell damage rely primarily on the salt-activated production of compatible solutes or penetrants [20,21] and the reduction of oxidative damage to protect membranes and proteins [22,23]. Chlorophyll fluorescence is an important parameter in studying the photosynthetic efficiency of plants [24], which directly reflects the response of plants to stress [25,26]. Therefore, the efficiency of photosystemic chemical reactions and the degree of heat dissipation under stress conditions can be estimated by measuring the chlorophyll fluorescence yield.

Developing new saline–alkali-tolerant rice germplasm is the most direct way but progress is limited because most genes exhibit traits and growth stages are specific [27]. It has been proven to be cheap and simple to improve plant salt tolerance by exogenous application of plant growth regulators [28,29,30]. As a new type of cyclohexane carboxylic acid plant growth retardant, prohexadione-calcium (Pro-Ca) has the effect of inhibiting vegetative growth, promoting reproductive growth, increasing yield, and improving quality [31]. Pro-Ca also has many applications in improving plant stress resistance. It also plays an important role in increasing tiller number, improving growth parameters, increasing photosynthetic pigment content, enhancing photosynthesis, membrane stability, and antioxidant capacity of rice under salt stress [32,33,34,35,36]. Pro-Ca can also improve the panicle traits of rice and increase yield by spraying at the booting stage under salt stress [37].

At present, the mechanism of salt tolerance of CMG varieties and how Pro-Ca can reduce yield loss by reducing the inhibition of salt stress on rice tillering are unclear. Therefore, this paper deeply studied the response mechanism of Pro-Ca to salt stress at the tillering stage and provided a reference for understanding Pro-Ca-induced plant resistance. The current study was intended: (A) to determine whether Pro-Ca can increase the yield of rice under NaCl stress; (B) to understand the process of Pro-Ca alleviating main and tiller damage at the tillering stage of rice under salt stress through phenotypic and physiological and biochemical levels. The results provide more ideas for understanding the adaptation mechanism of rice to salt stress and the mechanism of Pro-Ca, which is of great significance for the study of adversity stress and agricultural production.

## 2. Results

### 2.1. Effect of Pro-Ca on the Main Stem and Tiller Yield and Panicle Traits Under NaCl Stress

Compared to the Control, S treatment significantly reduced the number of tillers per plant (15.6%) and the total yield per plant (56.7%) (Figure 1A,D). S treatment reduced the number of effective panicles (9.09%) and increased the productive tillering rate of CMG by 6.45%, but not significantly (Figure 1B,C). Compared to S treatment, Pro-Ca + S treatment increased the number of tillers per plant (3.70%) and the number of effective panicles per plant (20.0%) (Figure 1A,B). Similarly, Pro-Ca + S treatment significantly increased the productive tillering rate and the total yield per plant by 17.1% and 59.4%, respectively (Figure 1C,D).

Compared to the Control, S treatment significantly reduced the panicle length, the number of primary branches, the number of secondary branches, the number of filled grains, and the 1000-grain weight of main stem and tillers of CMG variety (Table 1). The corresponding percent decrease of 16.3% and 15.7% in panicle length, 30.3% and 26.7% in the number of primary branches, 29.8% and 39.5% in the number of filled grains, and 33.8% and 5.22% in 1000-grain weight, respectively, for the main stem and tillers, and S treatment significantly reduced the number of secondary branches by 43.5% for the tillers. Interestingly, S treatment markedly improved the seed setting percentage by 18.0% and 23.2%, respectively, for main stem and tillers. Compared to S treatment, Pro-Ca + S treatment significantly increased the number of primary branches by 26.1%, the number of filled grains by 32.6%, and the 1000-grain weight by 39.5% in the main stem. Compared to S treatment, Pro-Ca + S treatment significantly increased the number of filled grains by 26.7% in tillers. Similarly, Pro-Ca + S treatment increased the 1000-grain weight by 1.24% in tillers under NaCl stress.

NaCl stress had a significant inhibitory effect on the yield of the main stem and tillers of CMG. Compared to the Control, S treatment significantly reduced the yield per plant of the main stem by 53.5% and the yield per plant of tillers by 57.5%. Compared with S treatment, Pro-Ca + S treatment significantly increased the yield per plant of main stem and tillers by 85.1% and 52.6%, respectively.

### 2.2. Effect of Pro-Ca on the Growth Relationship Between the Main Stem and Tiller Leaves at the Tillering Stage Under NaCl Stress

At the tillering stage, compared to the Control, S treatment significantly reduced the number of tillers, the number of leaves on the main stem, the number of tiller leaves at the fourth leaf axil, and the number of tiller leaves at the fifth leaf axil (Figure 2). The corresponding percent decrease of 33.3% in the number of tillers, 8.33% in the number of leaves on the main stem, 16.0% in the number of tiller leaves at the fourth leaf axil, and 83.3% in the number of tiller leaves at the fifth leaf axil, respectively (Figure 2A–C). Compared with S treatment, Pro-Ca + S treatment significantly increased the number of tillers by 100%, the number of main stem leaves by 5.45%, the number of tiller leaves at the fourth leaf axil by 14.3%, and the number of tiller leaves at the fifth leaf axil by 367%, respectively.

### 2.3. Effect of Pro-Ca on Morphological Characters of the Main Stem and the Tiller at the Fourth Leaf Axil Under NaCl Stress

Compared to the Control, S treatment significantly inhibited the plant height, stem diameter, and leaf area at the tillering stage of CMG (Table 2). The corresponding percent significantly decreased by 20.9% and 38.2% in plant height, 24.7% and 26.8% in stem diameter, and 58.3% and 66.4% in leaf area, respectively, for the main stem and the tiller at the fourth leaf axil. S treatment markedly reduced the root length by 28.8% in the tiller at the fourth leaf axil. Compared with S treatment, Pro-Ca + S treatment significantly increased the leaf area by 17.7% and 76.6% for the main stem and the tiller at the fourth leaf axil. Similarly, Pro-Ca + S treatment significantly increased the root length by 24.2% and stem diameter by 17.5% for the tiller at the fourth leaf axil.

Compared to the Control, S treatment significantly reduced the shoot dry weight, root dry weight, and seedling index of the main stem and the tiller at the fourth leaf axil (Table 3). The corresponding percent significantly decreased by 29.9% and 33.7% in shoot dry weight, 24.0% and 42.4% in root dry weight, and 24.7% and 38.7% in seedling index respectively for the main stem and the tiller at the fourth leaf axil. S treatment markedly improved the root–shoot radio (fresh weight) by 42.2% in the main stem and decreased it by 35.7% in the tiller at the fourth leaf axil. Compared to S treatment, Pro-Ca + S treatment significantly increased the root shoot ratio and seedling index of the main stem and the tiller at the fourth leaf axil. The corresponding percent significantly increased by 5.69% and 67.4% in root–shoot ratio, and 10.3% and 20.0% in seedling index respectively for the main stem and the tiller at the fourth leaf axil. Compared to S treatment, Pro-Ca + S treatment significantly increased the root dry weight by 5.00% for the main stem, and shoot dry weight by 6.30% for the tiller at the fourth leaf axil. Similarly, Pro-Ca + S treatment increased the shoot dry weight for the main stem by 2.77%, and the root dry weight for the tiller at the fourth leaf axil by 10.5%, but not significant.

### 2.4. Effect of Pro-Ca on Photosynthetic Pigment Content Under NaCl Stress

Compared to the Control, S treatment significantly reduced chl *a*, chl *b*, total chl, and carotenoid of CMG (Figure 3). The corresponding percent significantly decreased by 18.1% in chl *a*, 23.8% in chl *b*, 19.6% in total chl, and 21.9% in carotenoid for CMG. Compared to S treatment, Pro-Ca + S treatment markedly improved chl *a* by 32.9%, chl *b* by 58.4%, total chl content by 39.8%, and carotenoid content by 33.2%, respectively.

### 2.5. Effect of Pro-Ca on Gas Exchange Parameters Under NaCl Stress

Compared to the Control, S treatment significantly reduced the photosynthetic efficiency (Figure 4). The corresponding percent significantly decreased by 33.3% in *A*, 46.9% in *E*, 50.2% in *gsw*, 13.8% in *Ci*, and 22.3% in *AMC*, respectively (Figure 4A–D,F). S treatment significantly increased *Ls* by 29.2%, *WUEt* by 24.4%, and *WUEi* by 32.8% (Figure 4E,G,H). Compared to S treatment, Pro-Ca + S treatment markedly increased *A* by 25.0%, *E* by 71.8%, *gsw* by 69.4%, and *Ci* by 17.7%, respectively (Figure 4A–D). Similarly, Pro-Ca + S treatment increased *AMC* by 9.76% compared to S treatment (Figure 4F), but the difference was not statistically significant. Pro-Ca + S treatment significantly decreased *Ls* by 25.1%, *WUEt* by 28.2%, and *WUEi* by 27.2%, respectively, for CMG.

### 2.6. Effect of Pro-Ca on Chlorophyll Fluorescence Parameters Under NaCl Stress

Compared to the Control, NaCl stress significantly reduced Fm, Fv/Fm, ETR, Y(II), Fv/Fo, qP, and qL, respectively, of CMG variety (Figure 5A–G). The corresponding percent decrease of 18.8% in Fm, 5.50% in Fv/Fm, 9.19% in ETR, 8.98% in Y(II), 20.9% in Fv/Fo, 8.86% in qP, and 10.6% in qL, respectively, for CMG variety. Similarly, S treatment reduced qN^−1^ (0.80%) and Y(NO)^−1^ (0.55%) compared to the Control, but those differences were not statistically significant (Figure 5H,J). Furthermore, S treatment significantly increased Y(NPQ) by 5.13% (Figure 5K), and increased NPQ by 6.21% with no significance (Figure 5I). Compared to S treatment, Pro-Ca + S treatment markedly increased qN by 2.95%, NPQ by 9.00%, and Y(NPQ) by 5.32% for CMG, respectively (Figure 5H–J). Similarly, Pro-Ca + S treatment increased Fm by 3.01%, Fv/Fm by 0.69%, Fv/Fo by 2.76%, qP by 0.72%, and qL by 2.13%, respectively (Figure 5A,B,E–G). Compared to S treatment, Pro-Ca + S treatment reduced ETR by 2.83%, Y(II) by 3.02%, and Y(NO) by 4.19% (Figure 5C,D,J).

### 2.7. Pro-Ca Increased the Activity of Antioxidant Enzymes in Rice at the Tillering Stage Under NaCl Stress

Compared to the Control, NaCl stress significantly increased the activities of SOD by 19.7% and 8.48% and the activities of APX by 101% and 66.9%, respectively, for main stem and tiller leaves (Figure 6A,B,G,H). S treatment significantly decreased by 10.4% and 25.1% in the POD activity for main stem and tiller leaves (Figure 6C,D). Similarly, S treatment significantly decreased by 24.1% in the CAT activity for main stem leaves, and it decreased by 0.93% in the CAT activity for tiller leaves with no significance (Figure 6E,F). Compared to S treatment, Pro-Ca + S treatment significantly increased the activities of SOD by 7.47% and 14.7%, the activities of POD by 6.76% and 18.0%, and the activities of CAT by 26.4% and 58.5%, respectively, for main stem and tiller leaves (Figure 6A–F). Similarly, Pro-Ca + S treatment significantly increased the APX activity by 11.0% for the main stem leaves, and it increased the APX activity by 15.9% for tiller leaves with no significance (Figure 6G,H).

### 2.8. Effect of Pro-Ca on MDA Content Under NaCl Stress

Compared to the Control, S treatment markedly increased the MDA content by 59.2% and 61.4%, respectively, in main stem and tiller leaves (Figure 7A,B). Compared to S treatment, Pro-Ca + S treatment significantly reduced the MDA content^−1^ (10.3%) for the main stem leaves, and it reduced the MDA content^−1^ (2.19%) for tiller leaves with no significance.

### 2.9. Effect of Pro-Ca on Soluble Protein Content Under NaCl Stress

Compared to the Control, S treatment significantly decreased by 2.98% and 6.40% in soluble protein content, respectively, for the main stem and tiller leaves (Figure 8A,B). Compared to S treatment, Pro-Ca + S treatment markedly increased the soluble protein content by 2.60% and 6.08%, respectively, in main stem and tiller leaves.

## 3. Discussion

Effective tiller number is an important part of yield. Salt stress not only seriously affects the development and survival of primary and secondary tillers, but also leads to the increase in late abortion rate. In this study, NaCl stress significantly reduced the number of tillers per plant, the number of effective panicles, the percentage of tillers, and the total yield per plant at the maturity stage of CMG (Figure 1A–C), but increased the seed setting percentage of main stem and tillers (Table 1). This may be due to salt stress leading to reduced accumulation of assimilates during plant reproductive growth [38,39], poor panicle development, and reduced total grain number [40,41]. In contrast, the number of filled grains and 1000-grain weight were significantly increased after exogenous spraying of Pro-Ca, indicating that exogenous spraying of Pro-Ca may have the effect of expanding the flow of conductive tissues, accelerating the transport of assimilates, and expanding the storage capacity [42]. This is consistent with previous research results [37]. It is worth noting that in this study, the weakening effect of salt stress on the main stem of CMG 1000-grain weight was significantly stronger than that on tillers, while the application of Pro-Ca significantly increased the 1000-grain weight of main stem panicles under NaCl stress (Table 1). This result may be related to the accumulation of sodium ions in different parts [43], and Pro-Ca may be involved in the regulation of main stem and tiller assimilate transport, but further research is needed to prove this hypothesis.

At the same time, the increase in productive tillering rate at maturity was closely related to the development of the main stem and tiller at the tillering stage. Leaves are the main photosynthetic organs of crops. Leaf area is closely related to biomass and grain yield by increasing interception radiation and energy conversion capacity [44]. Salt stress hinders plant growth by limiting photosynthesis [45]. In our study, after 14 days of NaCl treatment, S treatment significantly reduced the tillering ability of CMG (Figure 2A), inhibited the growth of main stem leaves and tillering leaves at the corresponding leaf axils (Figure 2B,C), and led to the growth stagnation of high tiller meristems, at the same time, S treatment also reduced the total leaf area and seedling index of main stem and tiller (Table 2 and Table 3). Exogenous spraying of Pro-Ca significantly accelerated the growth of rice at low and high tillering positions, and increased the total photosynthetic area of main stem and tiller and seedling index under NaCl stress. Previous studies have also proved that Pro-Ca can promote the elongation of rice tillering buds under salt stress [33]. In summary, Pro-Ca can promote the growth of main stem and tiller leaves under NaCl stress, improve the leaf area and seedling index of tillers at low tillering positions, and provide a guarantee for increasing productive tillering rate at the maturity stage.

Biomass accumulation depends on the photosynthetic fixation of assimilates [46]. Chlorophyll content can reflect the development and photosynthetic capacity of chloroplasts. The significant decrease in chlorophyll concentration under salt stress indicated that chloroplasts were damaged by oxidation. In our study, NaCl stress resulted in a significant decrease in chl *a*, chl *b*, total chl content, and carotenoids (Figure 3A–D). The reason may be that salinity causes the obstruction of chlorophyll biosynthesis or the enhancement of chlorophyll enzyme activity [47]. However, spraying Pro-Ca significantly increased chl *a*, chl *b*, total chl content, and carotenoids in the rice tillering stage under NaCl stress. Carotenoids are involved in protecting chlorophyll pigments from ROS damage in chloroplasts [48]. Therefore, chloroplast repair may be related to the high level of carotenoid after spraying Pro-Ca. Previous studies have also shown that exogenous application of Pro-Ca can effectively repair the thylakoid structure in chloroplasts and provide a guarantee for the accumulation of assimilates under saline–alkali stress [35].

Photosynthesis is driven by light energy absorbed by chl pigments [49], and stomatal closure and reduced transpiration are the main mechanisms for plants to prevent leaf water potential from falling to dangerous levels [50]. This study shows that the main reason for the decrease in CMG photosynthetic activity under NaCl stress is the decrease in *gsw* and the increase in *Ls* (Figure 4C,E). It also further shows that under NaCl stress, plants can maintain cell hydration and prevent excessive water loss by controlling stomatal opening and closing [51]. The reason may be that NaCl stress leads to an increase in ABA content and induces stomatal closure, thus affecting photosynthetic efficiency [52]. In addition to the stomatal exchange of carbon dioxide, photosynthesis also depends on the transport of stomatal air space (mesophyll conduction) to carboxylase in the chloroplast mechanism [53]. Under NaCl stress, the mesophyll apparent conductance decreased significantly (Figure 4F), indicating that the decrease in photosynthetic rate in rice was also related to mesophyll limitation. Spraying Pro-Ca increased *A*, *gsw*, *Ci*, and *AMC* of CMG (Figure 4A,C,D,F), indicating that Pro-Ca may increase *A* by promoting the conversion of extracellular carbon dioxide to carboxylation sites under stomatal limitation caused by NaCl stress. This is consistent with the results of Pro-Ca improving salt tolerance of soybeans [35]. In addition, the decrease in *E* and the significant increase in *WUEt* and *WUEi* of S treatment (Figure 4B,G,H) indicated that the salt-tolerant variety CMG had better water regulation under NaCl stress. In summary, Pro-Ca can not only protect the photosynthetic structure of plant leaves under NaCl stress but also promote carbon dioxide assimilation and enhance photosynthetic activity under NaCl stress by regulating stomatal opening.

In addition to affecting the net assimilation of carbon dioxide, salt stress also leads to a decrease in the quantum yield of PSII under light conditions. Under normal conditions, the light energy captured by chlorophyll molecules is used for photochemical reactions to drive photosynthesis. If excessive, it is dissipated in the form of heat [49], that is, reversible damage of PSII is initiated by non-photochemical quenching to reduce excessive light energy and possible reactive oxygen species [54]. In this study, compared to the Control, the values of *A*, Fv/Fm, ETR, Y(II), qP, and qL were significantly reduced under NaCl stress (Figure 4A and Figure 5B–D,F,G), indicating the possible existence of persistent photoinhibition [55]. For salt-tolerant variety CMG, photosynthetic activity and light damage index Y(NO) of the two treatments subjected to NaCl stress was close to and slightly higher than that of the Pro-Ca treatments not subjected to NaCl stress (Figure 5C,D,F,G), it indicated that the light damage of salt-tolerant variety CMG was less. Similarly, the increases of qN, NPQ, and Y(NPQ) of the two stressed treatments (Figure 5H,I,K) indicate that the salt tolerance of CMG may be reflected in reducing the damage of salt to the photosynthetic system, which has also been proved by previous studies [8,9]. At the same time, the application of Pro-Ca significantly enhanced the photoprotective ability of CMG and reduced the photodamage. This is consistent with the results of previous studies on the significantly improved photoprotection ability of maize under chilling stress [56]. In summary, Pro-Ca can reduce the inhibition of PSII under NaCl stress by regulating the photoprotective response induced by excessive energy dissipation in PSII [57].

ROS is produced due to excessive photoexcitation pressure and excess capacity of PSII reaction center caused by stress [58]. The increase in SOD, POD, CAT, and APX activity can remove ROS in time to maintain normal metabolism [59]. Previous studies have shown that the activities of SOD, Cu/Zn-SOD, and APX in salt-tolerant rice varieties increase with the increase in salt concentration [60], which can also be proved in this study (Figure 6A,B,G,H). However, the activities of POD and CAT decreased significantly (Figure 6C,D,G,E), which may be related to the damage to plant enzyme protein structure caused by salt stress [61]. Exogenous spraying of Pro-Ca has a significant effect on improving the antioxidant enzyme activity of CMG main stem and tiller leaves under NaCl stress, which is consistent with previous research results [62]. These results indicate that exogenous application of Pro-Ca is involved in the regulation of the antioxidant system in rice under NaCl stress to protect the whole plant and reduce oxidative damage.

Excessive ROS produced by salt stress can change the metabolism and oxidative homeostasis of plant cells and produce membrane lipid peroxides [63]. MDA content as an indicator of membrane lipid peroxidation, reflecting the degree of damage to the leaves [64]. After the exogenous application of Pro-Ca, the MDA content of main stems and tillers decreased (Figure 6A,B), indicating that exogenous spraying of Pro-Ca could reduce cell membrane damage caused by NaCl stress. Salt stress can induce plant osmotic stress and lead to physiological drought by limiting root water absorption [65]. Exogenous spraying of Pro-Ca has a significant effect on increasing the soluble protein content of the main stem and tillers under NaCl stress (Figure 7A,B). It is also proved that Pro-Ca can increase the content of cell osmotic adjustment substances under NaCl stress, thereby increasing stomatal conductance and slowing down the inhibition of salt stress on photosynthetic efficiency [66].

## 4. Materials and Methods

### 4.1. Plant Materials and Test Conditions

Rice variety (*Oryza sativa* L.) CMG (salt-tolerant, landrace) was used as experimental material. Provided by the Germplasm Resource Library of Coastal Agricultural College of Guangdong Ocean University. In this experiment, the chemical reagent prohexadione-calcium (Pro-Ca) stock solution was 5%, which was provided by the Chemical Control Laboratory of Coastal Agricultural College of Guangdong Ocean University. The concentration of the regulator was 100 mg·L^−1^.

Morphological and physiological experiments: The experiment was conducted in the solar greenhouse of Guangdong Ocean University, Zhanjiang, China (21.27° N, 110.32° E) in August 2023. When the 4th true leaf appeared, rice seedlings were transplanted into the plastic pot (19 cm diameter × 15 cm bottom diameter × 18 cm height), with 3 kg of clay soil, with four plants per pot. The soil properties were as follows: soil organic carbon, 32.4 g·kg^−1^; available phosphorus, 4.0 mg·kg^−1^; available potassium, 48.4 mg·kg^−1^; alkali-hydrolyzable nitrogen, 37.1 mg·kg^−1^; and soil pH, 7.23.

A total of four treatments were set: (1) Control (water spraying + 0% NaCl), (2) Pro-Ca (100 mg·L^−1^ Pro-Ca leaf spraying + 0% NaCl), (3) S (water spraying + 0.3% NaCl), (4) Pro-Ca + S (100 mg·L^−1^ Pro-Ca leaf spraying + 0.3% NaCl). After regreening (6.1 leaf), the plant materials were divided into four groups before tillering, and two groups were sprayed with 100 mg·L^−1^ Pro-Ca on the leaves. Each leaf and sheath were sprayed evenly to the front and back without dripping. After 48 h of Pro-Ca treatment, NaCl solution was carried out twice, with an interval of 24 h each time, and a total of about 0.3% NaCl was added. The NaCl content of the water layer was detected by a hand-held SKD1688-TR-6 EC meter (Shunkeda Technology Co., Ltd., Beijing, China) to ensure that the salt content remained relatively stable. Samples were collected 14 days after NaCl treatment for determination. Before the experiment began, all plants were well grown (10 pots/treatment).

Yield test: The rice seedlings with 4th true leaf appeared were transplanted into black plastic barrels with specifications of 31.5 cm diameter × 22.5 cm bottom diameter × 29.5 cm height, 7 plants per barrel, and 15 kg of clay soil in barrel. The treatment time was consistent with the tillering stage test. The yield and panicle traits of rice were measured at the maturity stage. Before the experiment began, all plants were well grown (5 pots/treatment).

### 4.2. Yield and Panicle Traits

Plants were randomly selected from each treatment. The direct measurement method was used to determine the number of tillers: the number of effective panicles per plant, the number of primary branches, the number of secondary branches, the number of filled grains, the 1000-grain weight, and the seed setting percentage. The seed setting percentage is expressed by the ratio of the number of filled grains to the total number of grains per panicle. The productive tillering rate is the ratio of effective panicles per plant to tillers per plant. The total yield per plant is the sum of the main stem yield per plant and the total tiller yield per plant. Main stem yield per plant = total grain number of main stem panicle × seed setting percentage of main stem × grain weight of main stem; total tiller yield per plant = total grain number of tiller panicle × seed setting percentage of tiller × effective panicle number per plant × grain weight of tiller.

### 4.3. Morphological Parameters

After 14 days of salt stress, rice plants with consistent growth were harvested for morphological determination. The number of leaves with more than two leaves was recorded as a tiller, and the number of tillers was counted as a standard. The plant height and root length (0.1 cm) were measured by direct measurement method. The width of the stem base (0.1 mm) was measured by vernier caliper at the junction of the rhizome and rhizome. The leaf area (0.1 mm^2^) of survival leaves of the main stem and tillers was measured by leaf area meter (YX-1241). The plants were washed with deionized water and dried to determine the fresh weight of the aboveground and underground parts (0.0001 g). The samples were dried in a 70 °C oven for 72 h to constant weight to determine the dry weight of the aboveground and underground parts. Root–shoot ratio is the ratio of the fresh weight of the underground part to the aboveground part; the seedling index (SI) formula is as follows [67]:$$SI=\left(\frac{stem\;diameter}{plant\;height}+\frac{root\;dry\;weight}{shoot\;dry\;weight}\right)\times \left(root\;dry\;weight+shoot\;dry\;weight\right)$$

### 4.4. Photosynthetic Pigment Content

The photosynthetic pigment content was determined with a slight change from the Lichtenthaler’s method [68]. Chlorophyll *a* (chl *a*), chlorophyll *b* (chl *b*), total chlorophyll (chl *a* + *b*), and carotenoid content were determined by spectrophotometry after 0.02 g fresh leaves were soaked in 1.80 mL 95.0% ethanol in the dark until the leaves completely faded and whitened. The absorbance of the extract at 649, 665, and 470 nm wavelengths was determined by spectrophotometer (GENESYS 180 UV–Vis, Thermo Fisher Scientific, Waltham, MA, USA).$$\begin{array}{c} Chlorophyll\;a\;\left(chl\;a\right)=13.95OD_{665}-6.88OD_{649}\\ Chlorophyll\;b\;\left(chl\;b\right)=24.96OD_{649}-7.32OD_{665}\\ Total\;chlorophyll=chl\;a+chl\;b\\ Carotenoid=\left(1000OD_{470}-2.05\;chl\;a-111.48\;chl\;b\right)/245 \end{array}$$

### 4.5. Gas Exchange Parameters

The net photosynthetic rate (*A*), stomatal conductance (*gsw*), intercellular carbon dioxide concentration (*Ci*), and transpiration rate (*E*) of the fully expanded second leaf of rice were measured by Li-6800 portable photosynthesis system (Li-Cor, Lincoln, NE, USA). Five strains were selected for each treatment. After 14 days of NaCl treatment, 9:00–11:30 a.m. and 2:00–5:00 p.m. were performed. The relative humidity was set to 60–65%, the leaf temperature was set to 25 °C, the light intensity was set to 1000 μmol·m^−2^·s^−1^, the carbon dioxide concentration was set to 400 ± 5 μmol·mol^−1^, and the air flow rate was set to 500 μmol·s^−1^ [69]. The apparent mesophyll conductance (*AMC*) was expressed as the ratio of *A* to *Ci* [70]; the instantaneous water use efficiency (*WUEt*) was calculated by the ratio of *A* to *E*. The internal water use efficiency (*WUEi*) was calculated as the ratio of *A* to *gsw*.$$Ls=\left(1-\frac{Ci}{Ca}\right)\times 100\%$$

**Theorem** **1.**
*Ca is the ambient CO_2_ concentration, that is, the CO_2_ concentration at the instrument air intake.*


### 4.6. Chlorophyll Fluorescence

A portable modulated chlorophyll fluorometer (PAM-2500, Heinz Walz, Effeltrich, Germany) was used for in situ measurements of the second fully expanded leaves of rice. Prior to measurement, the plants were dark-adapted for 40 min. Then, the minimum chlorophyll fluorescence (Fo) and maximum chlorophyll fluorescence (Fm) were recorded. Following this, the light was turned ON and set to 334 μmol·m^−2^·s^−1^ (red light, with maximum emission at 630 nm), and the plants were allowed to adapt for 5 min to reach a steady state. The minimum fluorescence under light (Fo’), maximum fluorescence yield (Fm’), and steady-state fluorescence (F’) were then determined. The maximum quantum efficiency (Fv/Fm), actual quantum yield (YII), photochemical quenching coefficient (qP) and (qL), non-photochemical quenching coefficient (qN) and (NPQ), and non-photochemical quenching quantum yield Y(NO) and Y(NPQ) of photosystem II (PSII). Five strains were selected for each treatment.

### 4.7. Antioxidant Enzyme Activity

The fully expanded main stem and tiller leaves of rice were frozen in liquid nitrogen and then transferred to a −80 °C refrigerator for storage. After all samples were collected, they were determined. The leaves (0.5 g) were ground in liquid nitrogen, and then 10 mL of pre-cooled phosphate buffer solution (0.05 mmol·L^−1^ PBS, pH 7.8) was added. The homogenate was ground and centrifuged at 4 °C and 6000× *g* for 20 min. The activities of superoxide dismutase (SOD), catalase (CAT), peroxidase (POD), and ascorbate peroxidase (APX) in the supernatant were measured. Determination of antioxidant enzyme system: The activity of SOD was determined by the nitroblue tetrazolium (NBT) method [71]; the activity of CAT was determined by reference to the method of Aebi [72]; and POD activity was determined by guaiacol method [73]. The APX activity level was calculated according to the method described by Nakano and Asada [74].

### 4.8. Malondialdehyde Content

The content of malondialdehyde (MDA) was determined by the thiobarbituric acid (TBA) method [75]. The collected functional leaves (0.5 g) were ground in liquid nitrogen, then 10 mL of phosphate buffer solution (0.05 mmol·L^−1^ PBS, pH 7.8) was added, ground into homogenate, and centrifuged at 10,000 rpm and 4 °C for 10 min. Supernatant 1 mL was mixed with 2 mL of 0.6% thiobarbituric acid (TBA) in a centrifuge tube. The mixture was boiled in a boiling water bath for 15 min, and then centrifuged at 10,000 rpm and room temperature for 10 min. The absorbance of the supernatant was measured at 450 nm, 532 nm, and 600 nm using a spectrophotometer (GENESYS 180 UV–Vis, Thermo Fisher Scientific, Waltham, MA, USA).

### 4.9. Soluble Protein Content

The soluble protein was determined by Coomassie brilliant blue G-250 method according to Bradford’s method [76]. Next, 0.5 g sample was weighed in a pre-cooled mortar, 10 mL of 0.05 mol L^−1^ pre-cooled phosphate buffer (pH 7.8) was added in three times, the ice bath was ground into a homogenate, transferred to a centrifuge tube, centrifuged at 4 °C and 12,000 rpm for 2 min, and the supernatant was the crude protein extract. Then, 1 mL enzyme solution was added to 5 mL Coomassie brilliant blue solution and shaken well.

### 4.10. Statistical Analyses

All data were collected using Microsoft Excel 2019 software. Statistical analyses were performed using SPSS 26.0 (SPSS Corp., Chicago, IL, USA) and descriptive statistics were used to test the mean ± standard error of the mean (SEM). Statistical analysis, including one-way analysis of variance (ANOVA). Differences between means were evaluated by Duncan’s test at *p* < 0.05 significance level.

## 5. Conclusions

The results demonstrated that the application of Pro-Ca in pot experiments significantly enhanced various morphological parameters, including the growth relationship between the main stem and tiller leaves, leaf area, and seedling index under NaCl stress. Furthermore, it improved photosynthetic efficiency, photoprotection capacity, photosynthetic pigment concentration, antioxidant enzyme activity, osmotic adjustment compounds, membrane stability, as well as increased productive tillering rate, grain yield from both main stem and tillers, 1000-grain weight, and seed setting percentage. These findings underscore the critical role of alleviating salt-induced damage to tillers in enhancing rice yield under saline conditions. This experiment may have three main disadvantages: (A) It is based on a potted greenhouse experiment and has limitations to guide actual production. (B) There is only one study on the physiological metabolism of CMG under NaCl stress. To clarify the effect of Pro-Ca on improving plant salt tolerance, its dynamic changes should be continuously observed. (C) The effect of Pro-Ca on the carbon metabolism of rice under NaCl stress should be further studied. However, this study also provides partial insights into how plant growth regulators improve plant salt tolerance, helping to improve the ability of crops to withstand salt stress damage in coastal areas. Similarly, according to the results of our experiment, Pro-Ca not only has a significant effect on salt tolerance but also has a good effect on reducing plant height. This makes it possible to improve the plant’s ability to resist against lodging and improve grain production of rice.

## Figures and Tables

**Figure 1 plants-14-00188-f001:**
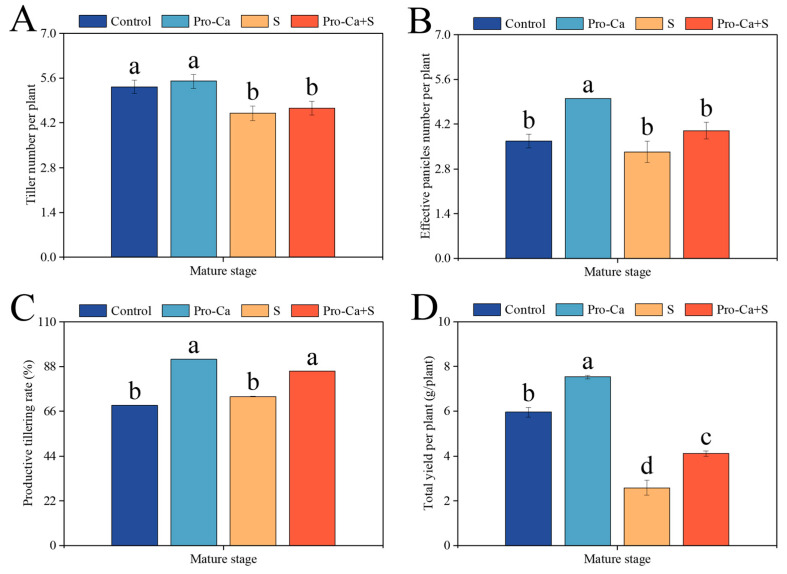
Effects of prohexadione-calcium (Pro-Ca) on the yield traits under NaCl stress. (**A**) Tiller number per plant, (**B**) effective panicles number per plant, (**C**) productive tillering rate, and (**D**) total yield per plant. Abbreviations: Water spraying + 0% NaCl (Control), 100 mg·L^−1^ Pro-Ca leaf spraying + 0% NaCl (Pro-Ca), water spraying + 0.3% NaCl (S), and 100 mg·L^−1^ Pro-Ca leaf spraying + 0.3% NaCl (Pro-Ca + S). Values described are the means ± SE (*n* = 6). Different letters denote significant differences (*p* < 0.05).

**Figure 2 plants-14-00188-f002:**
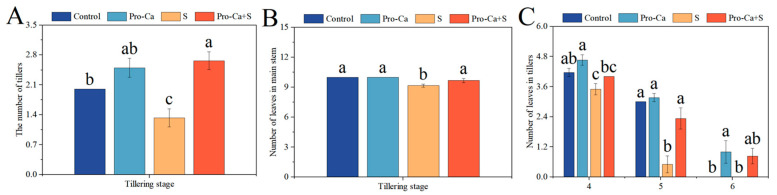
Effects of Pro-Ca on the growth relationship between the main stem and tiller leaves under NaCl stress. (**A**) The number of tillers, (**B**) number of leaves in the main stem, and (**C**) number of leaves in tillers, horizontal coordinates represent tiller position, vertical coordinates represent number of tiller leaves. Abbreviations: Water spraying + 0% NaCl (Control), 100 mg·L^−1^ Pro-Ca leaf spraying + 0% NaCl (Pro-Ca), water spraying + 0.3% NaCl (S), and 100 mg·L^−1^ Pro-Ca leaf spraying + 0.3% NaCl (Pro-Ca + S). Values described are the means ± SE (*n* = 6). Different letters denote significant differences (*p* < 0.05).

**Figure 3 plants-14-00188-f003:**
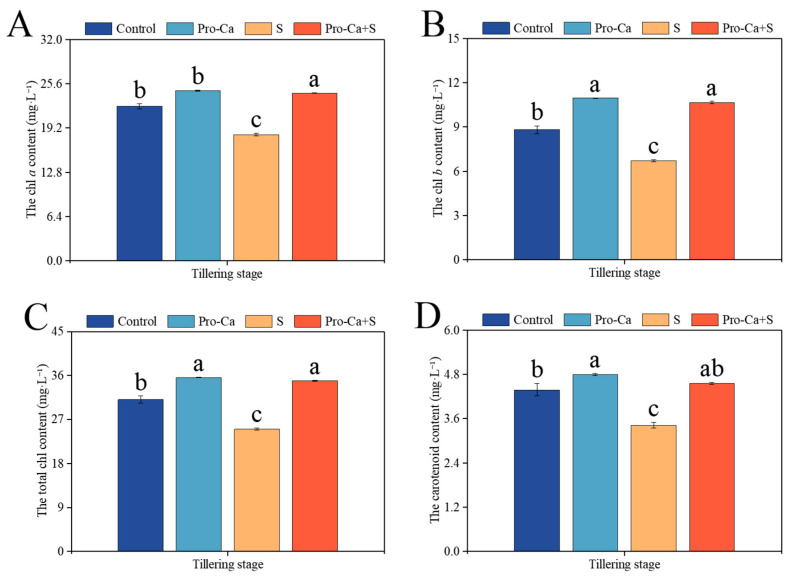
Effects of Pro-Ca on photosynthetic pigment content during the tillering stage under NaCl stress. (**A**) The chl *a* content, (**B**) the chl *b* content, (**C**) the total chl content, and (**D**) the carotenoid content. Abbreviations: Water spraying + 0% NaCl (Control), 100 mg·L^−1^ Pro-Ca leaf spraying + 0% NaCl (Pro-Ca), water spraying + 0.3% NaCl (S), and 100 mg·L^−1^ Pro-Ca leaf spraying + 0.3% NaCl (Pro-Ca + S). Values described are the means ± SE (*n* = 3). Different letters denote significant differences (*p* < 0.05).

**Figure 4 plants-14-00188-f004:**
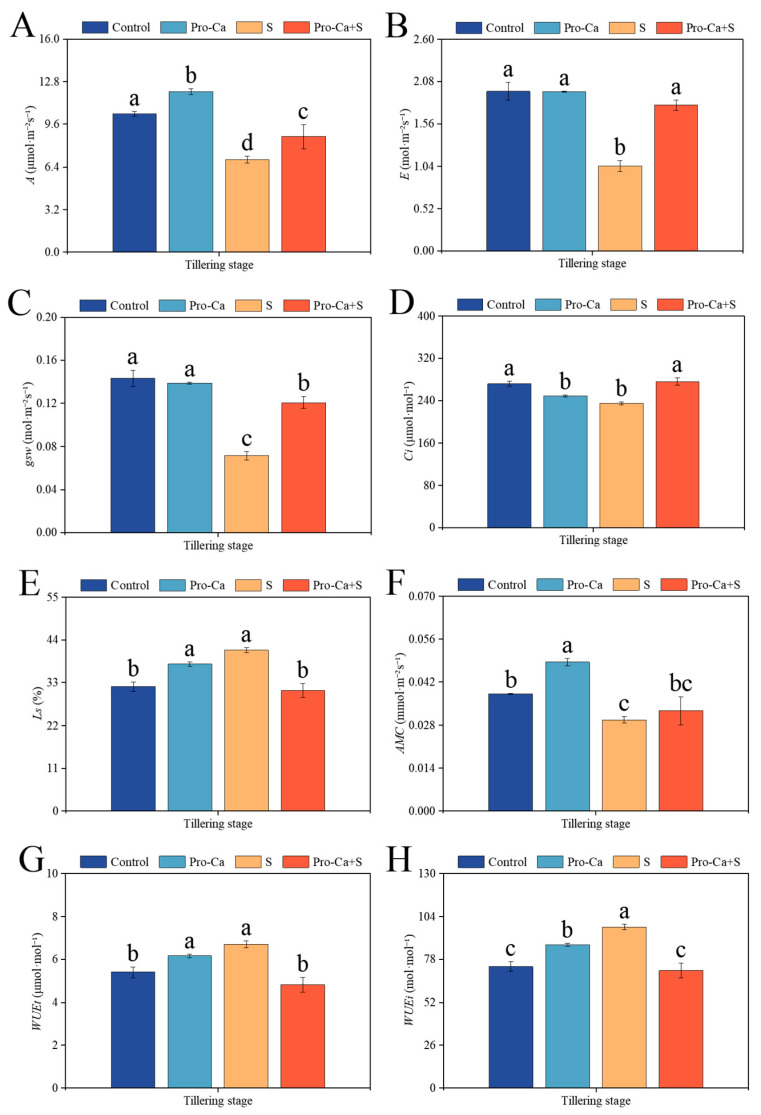
Effects of Pro-Ca on photosynthetic parameters of rice during the tillering stage under NaCl stress. (**A**) Net photosynthetic rate (*A*), (**B**) transpiration rate (*E*), (**C**) stomatal conductance (*gsw*), (**D**) intracellular CO_2_ concentrations (*Ci*), (**E**) stomatal limit (*Ls*), (**F**) apparent mesophyll conductance (*AMC*), (**G**) instantaneous water use efficiency (*WUEt*), and (**H**) internal water use efficiency (*WUEi*). Abbreviations: Water spraying + 0% NaCl (Control), 100 mg·L^−1^ Pro-Ca leaf spraying + 0% NaCl (Pro-Ca), water spraying + 0.3% NaCl (S), and 100 mg·L^−1^ Pro-Ca leaf spraying + 0.3% NaCl (Pro-Ca + S). Values described are the means ± SE (*n* = 3). Different letters denote significant differences (*p* < 0.05).

**Figure 5 plants-14-00188-f005:**
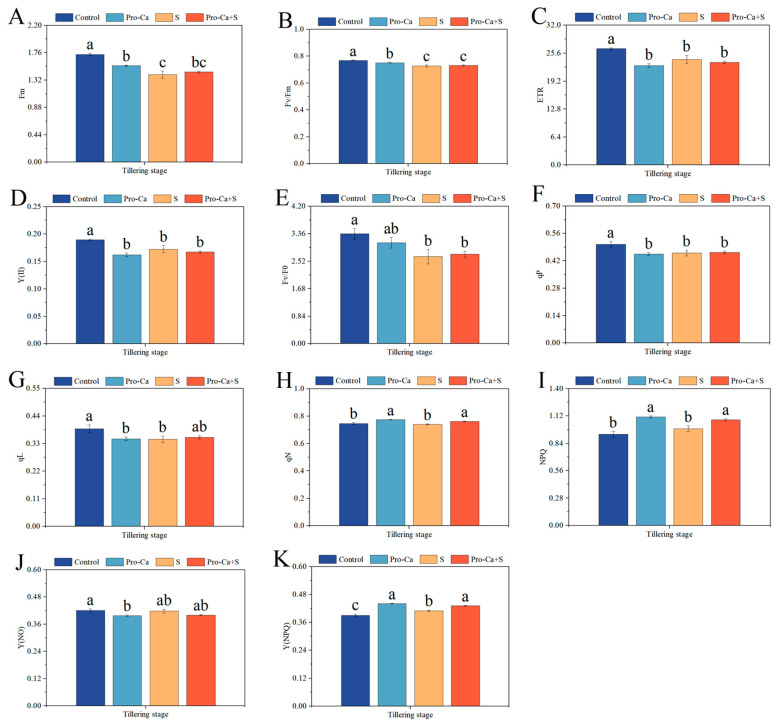
Effects of Pro-Ca on the chlorophyll fluorescence of rice during the tillering stage under NaCl stress. (**A**) Fm, (**B**) Fv/Fm, (**C**) ETR, (**D**) Y(II), (**E**) Fv/Fo, (**F**) qP, (**G**) qL, (**H**) qN, (**I**) NPQ, (**J**) Y(NO), and (**K**) Y(NPQ). Abbreviations: Water spraying + 0% NaCl (Control), 100 mg·L^−1^ Pro-Ca leaf spraying + 0% NaCl (Pro-Ca), water spraying + 0.3% NaCl (S), and 100 mg·L^−1^ Pro-Ca leaf spraying + 0.3% NaCl (Pro-Ca + S). Values described are the means ± SE (*n* = 3). Different letters denote significant differences (*p* < 0.05).

**Figure 6 plants-14-00188-f006:**
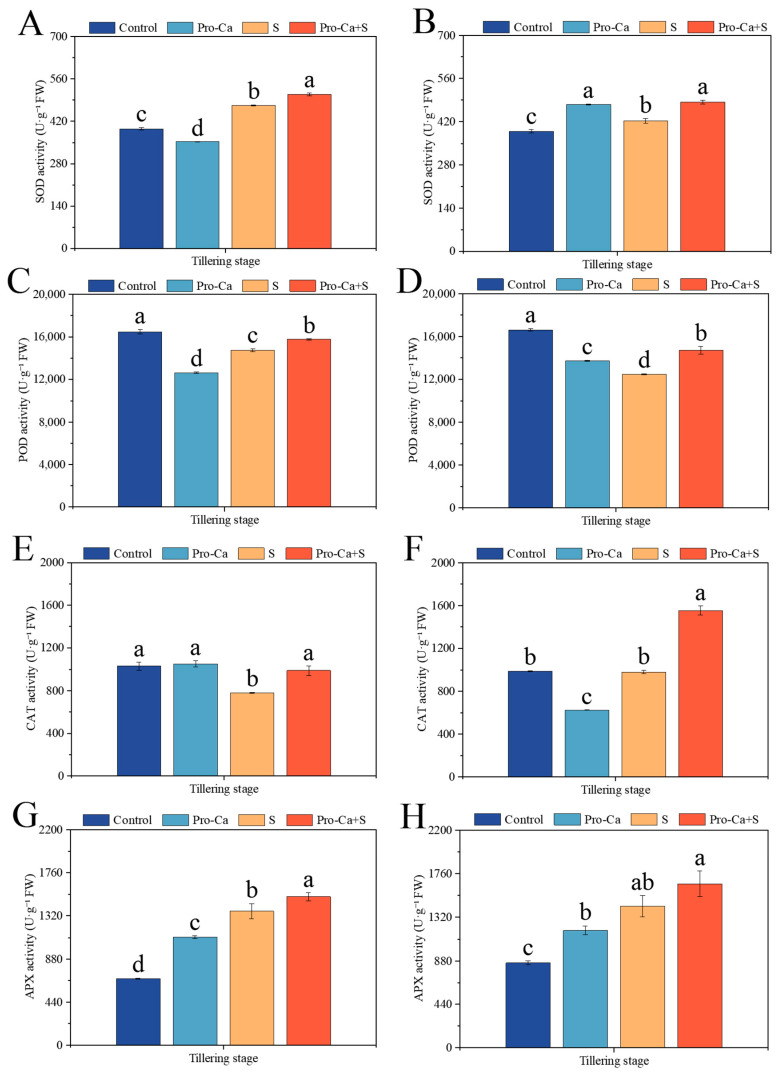
Effect of Pro-Ca on antioxidant enzyme activities of main stem and tiller of rice at tillering stage under NaCl stress. (**A**) The superoxide dismutase (SOD) of the main stem, (**B**) the superoxide dismutase (SOD) of tillers, (**C**) the peroxidase (POD) activity of the main stem, (**D**) the peroxidase (POD) activity of tillers, (**E**) the catalase (CAT) activity of the main stem, (**F**) the catalase (CAT) activity of tillers, (**G**) the ascorbate peroxidase (APX) activity of the main stem, and (**H**) the ascorbate peroxidase (APX) activity of tillers. Abbreviations: Water spraying + 0% NaCl (Control), 100 mg·L^−1^ Pro-Ca leaf spraying + 0% NaCl (Pro-Ca), water spraying + 0.3% NaCl (S), and 100 mg·L^−1^ Pro-Ca leaf spraying + 0.3% NaCl (Pro-Ca + S). Values described are the means ± SE (*n* = 3). Different letters denote significant differences (*p* < 0.05).

**Figure 7 plants-14-00188-f007:**
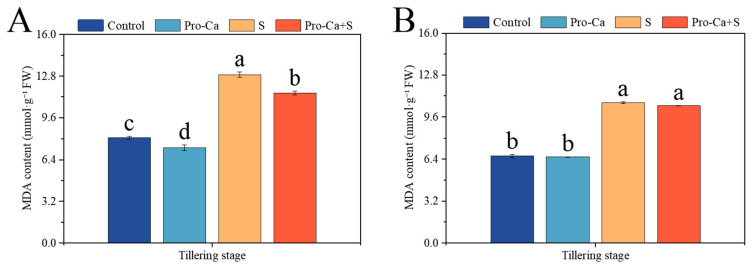
Effects of Pro-Ca on the membrane damage of rice during the tillering stage under NaCl stress. (**A**) Malondialdehyde (MDA) content of the main stem in CMG and (**B**) malondialdehyde (MDA) content of tillers in CMG. Abbreviations: Water spraying + 0% NaCl (Control), 100 mg·L^−1^ Pro-Ca leaf spraying + 0% NaCl (Pro-Ca), water spraying + 0.3% NaCl (S), and 100 mg·L^−1^ Pro-Ca leaf spraying + 0.3% NaCl (Pro-Ca + S). Values described are the means ± SE (*n* = 3). Different letters denote significant differences (*p* < 0.05).

**Figure 8 plants-14-00188-f008:**
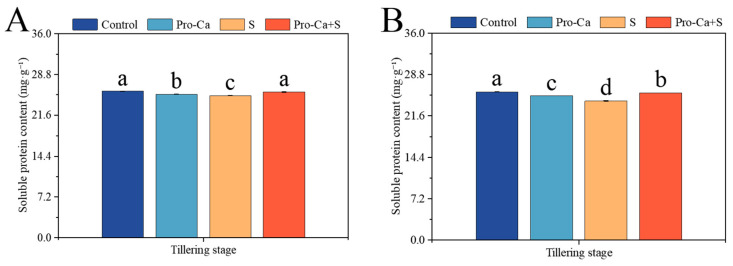
Effects of Pro-Ca on the soluble protein content of rice during the tillering stage under NaCl stress. (**A**) The soluble protein content of the main stem in CMG and (**B**) The soluble protein content of tillers in CMG. Abbreviations: Water spraying + 0% NaCl (Control), 100 mg·L^−1^ Pro-Ca leaf spraying + 0% NaCl (Pro-Ca), water spraying + 0.3% NaCl (S), and 100 mg·L^−1^ Pro-Ca leaf spraying + 0.3% NaCl (Pro-Ca + S). Values described are the means ± SE (*n* = 3). Different letters denote significant differences (*p* < 0.05).

**Table 1 plants-14-00188-t001:** Effects of Pro-Ca on panicle traits of the main stem and tiller in rice at maturity stage under NaCl stress.

Parts	Treatment	Panicle Length (cm)	Number of Primary Branches	Number of Secondary Branches	The Number of Filled Grains	1000-Grain Weight (g)	Seed Setting Percentage (%)	Yield (g·plant^−1^)
Main stem	Control	24.53 ± 0.18 a	11.00 ± 0.58 a	8.00 ± 0.58 a	43.67 ± 0.67 b	26.57 ± 0.04 b	54.25 ± 1.56 c	1.16 ± 0.02 b
	Pro-Ca	22.33 ± 0.49 b	10.00 ± 0.00 a	5.67 ± 0.33 b	52.67 ± 1.45 a	27.67 ± 0.06 a	81.05 ± 0.98 a	1.46 ± 0.04 a
	S	20.53 ± 0.44 c	7.67 ± 0.88 b	8.33 ± 0.88 a	30.67 ± 1.20 c	17.59 ± 0.24 d	64.01 ± 4.09 b	0.54 ± 0.02 d
	Pro-Ca + S	16.83 ± 0.33 d	9.67 ± 0.33 a	5.67 ± 0.33 b	40.67 ± 1.76 b	24.54 ± 0.33 c	65.24 ± 0.36 b	1.00 ± 0.06 c
Tiller	Control	22.87 ± 2.42 b	10.00 ± 0.00 a	7.67 ± 0.33 a	41.33 ± 1.20 b	25.56 ± 0.50 ab	54.85 ± 0.34 d	4.79 ± 0.21 b
	Pro-Ca	24.73 ± 0.09 a	8.33 ± 0.33 b	6.00 ± 0.00 b	50.33 ± 0.33 a	26.71 ± 0.10 a	69.93 ± 0.76 a	6.07 ± 0.05 a
	S	19.27 ± 0.64 c	7.33 ± 0.33 c	4.33 ± 0.33 c	25.00 ± 2.00 d	24.23 ± 0.52 c	67.58 ± 0.08 b	2.04 ± 0.32 d
	Pro-Ca + S	19.00 ± 0.06 c	7.00 ± 0.00 c	3.00 ± 0.00 d	31.67 ± 0.33 c	24.53 ± 0.22 bc	65.98 ± 0.39 c	3.11 ± 0.06 c

Abbreviations: Water spraying + 0% NaCl (Control), 100 mg·L^−1^ Pro-Ca leaf spraying + 0% NaCl (Pro-Ca), water spraying + 0.3% NaCl (S), and 100 mg·L^−1^ Pro-Ca leaf spraying + 0.3% NaCl (Pro-Ca + S). Values described are the means ± SE (*n* = 3). Different letters denote significant differences (*p* < 0.05).

**Table 2 plants-14-00188-t002:** Effect of Pro-Ca on morphological characters of the main stem and the tiller at the fourth leaf axil under NaCl stress.

Parts	Treatment	Plant Height (cm)	Root Length (cm)	Stem Diameter (mm)	Leaf Area (mm^2^)
Main stem	Control	97.00 ± 0.23 a	28.73 ± 0.49 b	7.13 ± 0.09 a	8972.97 ± 91.81 b
	Pro-Ca	76.40 ± 0.12 b	31.70 ± 0.23 a	6.93 ± 0.03 a	10,462.33 ± 80.74 a
	S	76.70 ± 0.06 b	28.53 ± 0.18 b	5.37 ± 0.15 b	3738.67 ± 28.77 d
	Pro-Ca + S	60.40 ± 0.06 c	26.70 ± 0.12 c	5.37 ± 0.09 b	4400.87 ± 13.91 c
Tiller	Control	79.91 ± 0.28 a	25.61 ± 0.06 a	5.23 ± 0.01 b	4132.58 ± 64.03 b
	Pro-Ca	65.52 ± 0.39 b	22.59 ± 0.09 b	5.73 ± 0.01 a	4928.43 ± 54.58 a
	S	49.44 ± 0.89 c	18.23 ± 0.69 c	3.83 ± 0.04 d	1389.23 ± 36.74 d
	Pro-Ca + S	46.21 ± 0.06 d	22.58 ± 0.10 b	4.50 ± 0.03 c	2453.49 ± 82.88 c

Abbreviations: Water spraying + 0% NaCl (Control), 100 mg·L^−1^ Pro-Ca leaf spraying + 0% NaCl (Pro-Ca), water spraying + 0.3% NaCl (S), and 100 mg·L^−1^ Pro-Ca leaf spraying + 0.3% NaCl (Pro-Ca + S). Values described are the means ± SE (*n* = 3). Different letters denote significant differences (*p* < 0.05).

**Table 3 plants-14-00188-t003:** Effect of Pro-Ca on morphological characters of the main stem and the lowest tiller under NaCl stress.

Parts	Treatment	Shoot Dry Weight (mg)	Root Dry Weight (mg)	Root Shoot Ratio (%)	Seedling Index (10^−2^)
Main stem	Control	874 ± 1 a	263 ± 6 a	40.8 ± 0.2 d	42.6 ± 1.1 b
	Pro-Ca	770 ± 10 b	273 ± 1 a	53.4 ± 0.2 c	46.4 ± 0.1 a
	S	613 ± 11 c	200 ± 2 c	58.0 ± 0.3 b	32.1 ± 0.5 d
	Pro-Ca + S	630 ± 13 c	210 ± 2 b	61.3 ± 0.2 a	35.4 ± 0.4 c
Tiller	Control	359 ± 5 a	66 ± 2 a	27.2 ± 1.0 ab	10.6 ± 0.4 a
	Pro-Ca	276 ± 5 b	51 ± 0 b	24.8 ± 0.3 b	8.9 ± 0.0 b
	S	238 ± 2 d	38 ± 3 c	17.5 ± 0.2 c	6.5 ± 0.5 d
	Pro-Ca + S	253 ± 2 c	42 ± 1 c	29.3 ± 1.3 a	7.8 ± 0.2 c

Abbreviations: Water spraying + 0% NaCl (Control), 100 mg·L^−1^ Pro-Ca leaf spraying + 0% NaCl (Pro-Ca), water spraying + 0.3% NaCl (S), and 100 mg·L^−1^ Pro-Ca leaf spraying + 0.3% NaCl (Pro-Ca + S). Values described are the means ± SE (*n* = 3). Different letters denote significant differences (*p* < 0.05).

## Data Availability

The datasets presented in this study are included in the main text; further inquiries can be directed to the corresponding author.

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
