# Peer review of "Prohexadione-Calcium Reduced Stem and Tiller Damage and Maintained Yield by Improving the Photosynthetic and Antioxidant Capacity of Rice (*Oryza sativa* L.) Under NaCl Stress"

_plants, 2025, doi:10.3390/plants14020188_

Round 1

Reviewer 1 Report

Comments and Suggestions for Authors

The article submitted for review is a very good study of the use of prohexadione calcium in the process of reducing abiotic stresses in plants, using rice as an example, which is one of the basic food products in the world. The studies focused on the assessment of the physiological properties of plants and their antioxidant potential. Overall, I highly value the manuscript. It was prepared with care and attention to detail.

The title and keywords are appropriate and reflect well the content of the work. The abstract is an appropriate summary of the assumptions and results obtained during the research conducted.

The introduction is sufficient. However, it is recommended to include information about rice. Other aspects discussed in the experiment (general background of the research, salinity, Pa-Co, need for innovation) are also discussed.

The results of the research are well described and appropriately presented in graphic form. In the description of the results, when citing specific numerical values, the rule of 3 significant digits should be followed (e.g. 1.23 or 12.3). This note also applies to the values given in the tables.

When describing figures, please mention individual elements after a comma or semicolon. This will help you better understand the text you are reading. In general, please pay attention to correct punctuation in the descriptions (this note applies to all table and figure captions).

In Table 1, please provide the unit of efficiency as a logarithm.

Graph 2C has no caption in the figure.

Standardize the way to cite figures throughout the text of the manuscript.

The discussion is well written and refers to both our own research results and the results obtained by other authors. I have no comments.

Materials and methods:

Was the chemical composition and pH of the soil used investigated?

Did the authors consider applying NaCl to the soil before spraying with the phytoprotective agent?

Measurement of Gas Exchange Parameters What made the authors choose this light intensity? In the literature, ranges from 300 to 1500 can be found; a commonly used light intensity is 300 μ mol m-2.

Chlorophyll fluorescence: Please provide the type of light, the peak wavelength of the source, and the maximum light intensity. Yield and panicle characteristics should be placed next to the morphological assessment.

The consistency of the methods should be consistent with the order of presentation of the research results.

The conclusions are well-constructed and reflect the idea of the study. However, a sentence should be added stating the possible use of the results of the research conducted in agricultural practice.

Author Response

Comments 1: The title and keywords are appropriate and reflect well the content of the work. The abstract is an appropriate summary of the assumptions and results obtained during the research conducted.

Response 1: Thank you for pointing this out. We agree with this comment. We have made changes to the summary section. Please see the manuscript for details (Lines 19-22).

Comments 2: The introduction is sufficient. However, it is recommended to include information about rice. Other aspects discussed in the experiment (general background of the research, salinity, Pa-Co, need for innovation) are also discussed.

Response 2: Thanks for valuable suggestions. We have added other aspects discussed in the trial (general background of the research, salinity, Pa-Ca, need for innovation). Please see the manuscript for details (Lines 49-50).

Comments 3: The results of the research are well described and appropriately presented in graphic form. In the description of the results, when citing specific numerical values, the rule of 3 significant digits should be followed (e.g. 1.23 or 12.3). This note also applies to the values given in the tables.

Response 3: Thanks for valuable suggestions. In the result descriptions and tables, we have standardized on 3 significant digits. Please see the manuscript for details (e.g. Line 102).

Comments 4: When describing figures, please mention individual elements after a comma or semicolon. This will help you better understand the text you are reading. In general, please pay attention to correct punctuation in the descriptions (this note applies to all table and figure captions).

Response 4: Thank you for pointing out the errors. We have changed the incorrect punctuation in the description and title of the charts. Please see the manuscript for details (e.g. Lines 110-111).

Comments 5: In Table 1, please provide the unit of efficiency as a logarithm.

Response 5: Thank you for pointing out the errors. We have changed the efficiency units in Table 1 to logarithmic form. Please see the manuscript for details (Lines 135-136).

Comments 6: Graph 2C has no caption in the figure.

Response 6: Thank you for pointing out the errors. Caption of figure 2C has been added at the appropriate place in the manuscript. Please see the manuscript for details (Lines 153-154).

Comments 7: Standardize the way to cite figures throughout the text of the manuscript.

Response 7: Thanks for valuable suggestions. I have standardized the way to cite figures throughout the text of the manuscript. Please see the manuscript for details (e.g. Line 115).

Comments 8: Was the chemical composition and pH of the soil used investigated?

Response 8: Yes. Thanks for valuable suggestions. I have added the data of chemical composition and pH of the soil to the manuscript. Please see the manuscript for details (Lines 422-424).

Comments 9: Did the authors consider applying NaCl to the soil before spraying with the phytoprotective agent?

Response 9: Thanks for valuable suggestions. According to Zhao et al (Regulatory effects of Hemin on prevention and rescue of salt stress in rapeseed (Brassica napus L.) seedlings) (https://doi.org/10.1186/s12870-023-04595-z), phytoprotective agent would be more effective if sprayed before NaCl application, so our experimental design chose to apply NaCl solution after spraying Pro-Ca. The application of NaCl prior to the spraying of plant protectants is also a good experimental design that deserves continued research.

Comments 10: Measurement of Gas Exchange Parameters What made the authors choose this light intensity? In the literature, ranges from 300 to 1500 can be found; a commonly used light intensity is 300 μ mol m-2.

Response 10: Thank you for your valuable suggestions. A light intensity of 1000 μmol·m⁻²·s⁻¹ is close to the saturation point for rice plants. This is why we chose 1000 μmol·m⁻²·s⁻¹. The reference has also been added (Line 483).

Comments 11: Chlorophyll fluorescence: Please provide the type of light, the peak wavelength of the source, and the maximum light intensity. Yield and panicle characteristics should be placed next to the morphological assessment.

Response 11: Thanks for valuable suggestions. We have added the type of light, the peak wavelength of the source, and the maximum light intensity regarding chlorophyll fluorescence to the manuscript Materials and Methods, and adjusted the position of yield and panicle characteristics in the study results. Please see the manuscript for details (Lines 489-496 and lines 443-453).

Comments 12: The consistency of the methods should be consistent with the order of presentation of the research results.

Response 12: Thank you for pointing out the errors. We have changed the experimental methodology and the order in which the findings are presented and made them consistent. Please see the manuscript for details (Line 442 and line 100).

Comments 13: The conclusions are well-constructed and reflect the idea of the study. However, a sentence should be added stating the possible use of the results of the research conducted in agricultural practice.

Response 13: Thanks for valuable suggestions. We have summarized the uses of Pro-Ca in agricultural practices based on the results of our trials. Please see the manuscript for details (Lines 553-556).

Reviewer 2 Report

Comments and Suggestions for Authors

This study is of significance for analyzing the physiological mechanisms of rice adaptation to salt stress. The experimental design was reasonable, experimental methods were correct, and data collection was comprehensive. However, there are still some issues with the manuscript that require revisions before it can be accepted. If the author can make modifications in the following aspects, it will help with the acceptance of the manuscript.

Line 422-432:According to the text, the experimental materials were divided into 4 groups, with 2 groups subjected to Pro-Ca treatment. After 48 hours of the Pro-Ca treatment, NaCl treatment was performed. This may lead to the misconception that these two sets of test materials have been treated with both Pro-Ca and NaCl. In fact, it's not like that. It should be explained first that a total of 4 treatments were set, and then how Pro-Ca treatment and NaCl treatment were carried out separately.

According to the text, 12 plants were selected for the determination of morphological traits (Line 440), and 10 plants were selected for the determination of yield and panicle traits (Line 514). But why is the average value of data in Figures 1 and 2 n=6, and the average value of data in Tables 1, 2, and 3 n=3?

In paragraph 1 of Results 2.1, sentence 1 was doubled with sentence 2. In sentence 2, “S treatment significantly reduced the number of effective panicles” was not inconsistent with the results of Figure 1. According to Figure 1B, NaCl treatment reduced the number of effective panicles, but not significant. Furthermore, in this paragraph, it should be better to represent all the “plant-1” with “per plant”.

In line 113-115, the statement “Compared to the control, S treatment significantly reduced the panicle length, the number of primary branches, the number of secondary branches, the number of filled grains, and the 1000-grain weight of main stem and tillers of CMG variety” was not inconsistent with the results of Table 1. According to Table 1, the number of secondary branches of the main stem (8.33) was increased compared to the control (8.00). The percentage decrease in each trait compared to the control described in the manuscript is incorrect. Please carefully verify. For example, under salt stress, the panicle length of the main stem was 20.52cm, while the control was 24.53cm, a decrease of 16.31%, while the manuscript showed a decrease of 33.78%.

There are two citation formats for figures in the main text: “Figure” and “Fig.”, which should be consistent. Some citation formats have errors, such as in line 101, “(Figure 1A, 1B, D)” should be “(Figure 1A, B, D)”.

Author Response

Comments 1: Line 422-432:According to the text, the experimental materials were divided into 4 groups, with 2 groups subjected to Pro-Ca treatment. After 48 hours of the Pro-Ca treatment, NaCl treatment was performed. This may lead to the misconception that these two sets of test materials have been treated with both Pro-Ca and NaCl. In fact, it's not like that. It should be explained first that a total of 4 treatments were set, and then how Pro-Ca treatment and NaCl treatment were carried out separately.

Response 1: Thanks for valuable suggestions. This error has been corrected in the manuscript. Please see the manuscript for details (Lines 425-435).

Comments 2: According to the text, 12 plants were selected for the determination of morphological traits (Line 440), and 10 plants were selected for the determination of yield and panicle traits (Line 514). But why is the average value of data in Figures 1 and 2 n=6, and the average value of data in Tables 1, 2, and 3 n=3?

Response 2: We have made changes in the text. In addition, n = 3 and n = 6 represent 3 biological replicates and 6 biological replicates, respectively. I think yield and panicle characteristics need more data to prove the value of Pro-Ca in production (Line 443 and lines 455-456).

Comments 3: In paragraph 1 of Results 2.1, sentence 1 was doubled with sentence 2. In sentence 2, “S treatment significantly reduced the number of effective panicles” was not inconsistent with the results of Figure 1. According to Figure 1B, NaCl treatment reduced the number of effective panicles, but not significant. Furthermore, in this paragraph, it should be better to represent all the “plant-1” with “per plant”.

Response 3: Thank you for pointing this out. We have corrected this error in the manuscript (Lines 101-108).

Comments 4: In line 113-115, the statement “Compared to the control, S treatment significantly reduced the panicle length, the number of primary branches, the number of secondary branches, the number of filled grains, and the 1000-grain weight of main stem and tillers of CMG variety” was not inconsistent with the results of Table 1. According to Table 1, the number of secondary branches of the main stem (8.33) was increased compared to the control (8.00). The percentage decrease in each trait compared to the control described in the manuscript is incorrect. Please carefully verify. For example, under salt stress, the panicle length of the main stem was 20.52cm, while the control was 24.53cm, a decrease of 16.31%, while the manuscript showed a decrease of 33.78%.

Response 4: Thank you for pointing out the errors. We have corrected this error in the manuscript (Lines 115-128).

Comments 5: There are two citation formats for figures in the main text: “Figure” and “Fig.”, which should be consistent. Some citation formats have errors, such as in line 101, “(Figure 1A, 1B, D)” should be “(Figure 1A, B, D)”.

Response 5: Thank you for pointing out the errors. We have corrected this error in the manuscript (e.g. Lines 107-108 and line 147).
